# Switch from Omalizumab to Benralizumab in Allergic Patients with Severe Eosinophilic Asthma: A Real-Life Experience from Southern Italy

**DOI:** 10.3390/biomedicines9121822

**Published:** 2021-12-03

**Authors:** Corrado Pelaia, Claudia Crimi, Santi Nolasco, Giovanna Elisiana Carpagnano, Raffaele Brancaccio, Enrico Buonamico, Raffaele Campisi, Claudia Gagliani, Vincenzo Patella, Girolamo Pelaia, Giuseppe Valenti, Nunzio Crimi

**Affiliations:** 1Department of Health Sciences, University “Magna Graecia” of Catanzaro, 88100 Catanzaro, Italy; pelaia.corrado@gmail.com; 2Department of Clinical and Experimental Medicine, University of Catania, 95123 Catania, Italy; dott.claudiacrimi@gmail.com (C.C.); nolascos@hotmail.it (S.N.); raffaelemd@hotmail.it (R.C.); crimi@unict.it (N.C.); 3Department of Basic Medical Sciences, Neuroscience and Sense Organs, Section of Respiratory Disease, University “Aldo Moro” of Bari, 70121 Bari, Italy; elisiana.carpagnano@uniba.it (G.E.C.); enricobuonamico@gmail.com (E.B.); 4Division of Allergy and Clinical Immunology, Department of Medicine, “Santa Maria della Speranza” Hospital, 84091 Battipaglia, Italy; raffaele.brancaccio@gmail.com (R.B.); vpatella@tiscali.it (V.P.); 5Allergology and Pulmonology Unit, Provincial Outpatient Center of Palermo, 90100 Palermo, Italy; gaglianiclaudia@yahoo.it (C.G.); vasvalenti@gmail.com (G.V.)

**Keywords:** severe asthma, allergy, eosinophils, omalizumab, benralizumab, switching

## Abstract

Background. The wide availability of monoclonal antibodies for the add-on therapy of severe asthma currently allows for the personalization of biologic treatment by selecting the most appropriate drug for each patient. However, subjects with overlapping allergic and eosinophilic phenotypes can be often eligible to more than one biologic, so that the first pharmacologic choice can be quite challenging for clinicians. Within such a context, the aim of our real-life investigation was to verify whether allergic patients with severe eosinophilic asthma, not adequately controlled by an initial biologic treatment with omalizumab, could experience better therapeutic results from a pharmacologic shift to benralizumab. Patients and methods. Twenty allergic patients with severe eosinophilic asthma, unsuccessfully treated with omalizumab and then switched to benralizumab, were assessed for at least 1 year in order to detect eventual changes in disease exacerbations, symptom control, oral corticosteroid intake, lung function, and blood eosinophils. Results. In comparison to the previous omalizumab therapy, after 1 year of treatment with benralizumab our patients experienced significant improvements in asthma exacerbation rate (*p* < 0.01), rescue medication need (*p* < 0.001), asthma control test (ACT) score (*p* < 0.05), forced expiratory volume in the first second (FEV_1_) (*p* < 0.05), and blood eosinophil count (*p* < 0.0001). Furthermore, with respect to the end of omalizumab treatment, the score of sino-nasal outcome test-22 (SNOT-22) significantly decreased after therapy with benralizumab (*p* < 0.05). Conclusion. The results of this real-life study suggest that the pharmacologic shift from omalizumab to benralizumab can be a valuable therapeutic approach for allergic patients with severe eosinophilic asthma, not adequately controlled by anti-IgE treatment.

## 1. Introduction

Severe asthma includes several phenotypes driven by complex cellular and molecular pathogenic networks defined as endotypes [1,2,3]. In particular, the phenotypic/endotypic characterization of asthma is mainly based on the distinction between type 2 (T2-high) and non-type 2 (T2-low) immune-inflammatory traits [4,5,6]. Type 2 asthma is predominantly sustained by bronchial eosinophilic inflammation, induced by either allergic or non-allergic mechanisms [7,8]. Actually, most cases of severe asthma are underpinned by airway eosinophilia [9,10]. The key cells implicated in the onset, progression, and worsening of type 2 asthma are T helper 2 (Th2) lymphocytes and group 2 innate lymphoid cells (ILC2), which release high amounts of interleukins (IL)-4 (IL-4), -13 (IL-13), and -5 (IL-5) [11,12,13]. IL-4 and IL-13 promote the synthesis of immunoglobulins E (IgE) [14]. IL-13 also induces goblet cell metaplasia, airway hyperresponsiveness, and proliferation of bronchial smooth muscle cells [15,16]. Moreover, IL-13 enhances the expression of the inducible isoform of nitric oxide synthase (iNOS), thereby stimulating the airway epithelium to produce nitric oxide (NO), with a consequent increase of fractional NO in exhaled breath (FeNO) [17]. IL-5 is essential for maturation, survival, and degranulation of eosinophils [18].

Hence, IgE, blood eosinophils and FeNO are currently considered as useful biomarkers of type 2 asthma, being also utilized to guide the choice of the most appropriate biological therapy for severe disease [1]. Indeed, IgE-driven positivity of skin prick test (SPT) to perennial allergens favours the prescription of anti-IgE treatment, whereas high blood eosinophil counts predict a good therapeutic response to either anti-IL-5 or anti-IL-5-receptor drugs, and high FeNO levels suggest the recommendation of a dual IL-4/IL-13 receptor blocker [19,20,21]. Despite such a quite schematic scenario, in clinical practice many patients with severe asthma can be eligible to more than one biologic because they express overlapping phenotypes [21,22,23]. For example, allergy and eosinophilic inflammation often coexist in the same subject, thus making him/her potentially responsive to either omalizumab, mepolizumab, or benralizumab [8,24].

Omalizumab is a humanized monoclonal antibody which specifically binds to the two Cε3 domains of the constant region of human IgE, thus preventing their interactions with both high affinity (FcεRI) and low affinity (FcεRII/CD23) IgE receptors [25,26]. Therefore, omalizumab effectively inhibits all IgE-dependent pathogenic effects occurring at level of immune/inflammatory and structural cells of the airways [26]. Both randomized controlled trials (RCT) and real-life clinical studies have clearly shown that omalizumab is very effective in improving several clinical and functional parameters related to severe asthma, thus positively impacting on disease exacerbations and airflow limitation [26,27,28,29,30,31,32]. A recent real-life observation has also shown that omalizumab can be very useful to control severe allergic asthma even during the COVID-19 pandemic [33]. However, some allergic patients with severe asthma who are partially or completely unresponsive to omalizumab may benefit from a therapeutic switching to mepolizumab [34,35,36,37,38].

Mepolizumab is a humanized IgG1/κ monoclonal antibody which selectively ligates IL-5, thus inhibiting its interaction with the α subunit of the IL-5 receptor (IL-5Rα) [39]. Both RCT and real-world experiences, carried out in patients with severe eosinophilic asthma, demonstrated that mepolizumab successfully decreases asthma exacerbations and blood eosinophils, as well as improves lung function and reduces the intake of oral corticosteroids (OCS) [39,40,41]. Moreover, in real life, mepolizumab has been shown to exert similar therapeutic effects in subjects with either allergic or non-allergic severe eosinophilic asthma [42].

Differently from mepolizumab, the humanized IgG1/κ monoclonal antibody benralizumab specifically binds through the Fab fragments to IL-5Rα, thereby impeding its occupancy by the natural ligand IL-5 [43,44,45]. Additionally, via the constant Fc portion benralizumab interacts with the FcγRIIIa receptor expressed by natural killer (NK) cells, thus promoting the release of proapoptotic proteins such as granzymes and perforins, which induce eosinophil apoptosis by the so-called mechanism of antibody-dependent cell-mediated cytotoxicity (ADCC) [43,44,45]. Both RCT and real-life reports have documented that benralizumab, utilized in subjects with either allergic or non-allergic severe eosinophilic asthma, significantly improves symptom control, disease exacerbations, lung function, OCS dependence, and blood eosinophilia [43,44,45,46,47,48,49,50,51]. However, the possibility of a therapeutic shift from omalizumab to benralizumab has been poorly investigated so far.

Therefore, on the basis of the above considerations, we decided to evaluate, within a real-life context, the potential benefits of a therapeutic shift to benralizumab, implemented in allergic patients with severe eosinophilic asthma, not satisfactorily controlled by a previous treatment with omalizumab.

## 2. Patients and Methods

### 2.1. Study Design and Endpoints

We performed a real-life, retrospective, multicenter study including adult outpatients (≥18 years old) with severe persistent allergic and eosinophilic asthma, uncontrolled despite the add-on biological treatment with omalizumab, and thus switched to benralizumab. These subjects referred to five Respiratory and Allergology Units of Southern Italy (Bari, Battipaglia, Catania, Catanzaro, and Palermo). All enrolled patients met the European Respiratory Society (ERS)/American Thoracic Society (ATS) criteria that define severe uncontrolled asthma [52].

At baseline (before any biological therapy), all 20 patients, also including 9 subjects complaining of chronic rhinosinusitis with nasal polyps (CRSwNP), exhibited high blood eosinophil counts and experienced frequent asthma exacerbations and an inadequate symptom control, thus needing high dosages of inhaled corticosteroids (ICS)/long-acting β_2_-adrenergic agonists (LABA) combinations, eventually associated with long-acting muscarinic antagonists (LAMA). Such patients had total serum IgE levels included within the range of 30–1500 IU/mL, as required by the Italian Drug Agency (AIFA), and they were also characterized by SPT positivity versus perennial aeroallergens. Therefore, omalizumab was prescribed and administered every 2 or 4 weeks, according to body weight and serum IgE levels. After a treatment period of at least 12 months, patients were shifted to benralizumab because of their unsatisfactory clinical response to anti-IgE therapy. We chose such a relatively long period of initial treatment, because it has been interestingly observed that the onset of a late response to omalizumab can be experienced by some patients even 48 weeks after starting anti-IgE therapy [53]. Therefore, a clinical evaluation restricted to only 3–6 months carries the risk of considering as a non-responder a potential late-responder to omalizumab [53]. Omalizumab failure was defined after at least 12 months of add-on therapy as lack of effectiveness, featured by recurrent asthma exacerbations and/or uncontrolled symptoms. Indeed, a clinically satisfactory response to any biological treatment for severe asthma should primarily provide an almost complete prevention of disease exacerbations. Because many severe asthmatic patients complain of very frequent exacerbations, even halving these events is often not enough to improve the overall quality of life. Benralizumab was prescribed according to current eligibility indications, with a regimen schedule consisting of a single subcutaneous injection of 30 mg every 4 weeks for the first three times, followed by the same dosage given every 8 weeks thereafter. Each subject had blood eosinophil levels higher than 300 cells/μL before starting benralizumab.

The main aim of this real-life investigation was to evaluate the effects of benralizumab on clinical, functional, and laboratory parameters in patients with severe type 2 asthma previously treated with omalizumab. The numbers of asthma exacerbations and hospitalizations, the use of short-acting β_2_-agonist (SABA) rescue medications, as well as prednisone intake, asthma control test (ACT) score, forced expiratory volume in the first second (FEV_1_), and blood eosinophil counts were recorded at baseline, after at least 12 months of omalizumab therapy (pre-shift), and at least 12 months after the first injection of benralizumab. Moreover, the sino-nasal outcome test-22 (SNOT-22) questionnaire was administered to patients with allergic CRSwNP. SPT was judged to be positive when the wheal area was equal to or larger than that one caused by histamine. Spirometry was performed according to ATS/ERS guidelines [54].

We enrolled 20 patients, including 13 females and 7 males, with a mean age of 52.85 ± 9.39 years, and a mean BMI of 23.47 ± 3.54 Kg/m^2^. The mean age of asthma onset was 31.40 ± 14.34 years. Mean baseline FEV_1_ was 62.47 ± 13.84% of predicted value. Baseline patient characteristics are summarized in Table 1. Fifteen patients (75%) were using LAMA, and 14 (70%) were taking leukotriene receptor antagonists (LTRA). With regard to asthma comorbidities, 11 subjects (55%) had gastro-esophageal reflux disease (GERD), 9 (45%) suffered from CRSwNP, 7 (35%) had bronchiectasis, 3 (15%) complained of atopic dermatitis, and 2 (10%) suffered from obstructive sleep apnea syndrome (OSAS).

The present observational study met the standards of Good Clinical Practice (GCP) and the principles of the Declaration of Helsinki. All recruited patients signed a written informed consent. Our study was also carried out in agreement to what decided by the local Ethical Committee of Calabria Region (Catanzaro, Italy; document n. 257—15 July 2021).

### 2.2. Statistical Analysis

Prism Version 9.1.2 (GraphPad Software Inc., San Diego, CA, USA) was used to statistically analyze study results. Median values with interquartile range (IQR) were used to express skewed data distributions, while normally distributed data were expressed as mean ± standard deviation (SD). The normality of data distribution was checked using Anderson–Darling and Kolmogorov–Smirnov tests. Dunnett’s multiple comparison test and Friedman test were used to compare variables, when appropriate. A *p*-value lower than 0.05 (two-sided) was considered to be statistically significant.

## 3. Results

After treatment with benralizumab, all patients reported a considerable reduction of asthma exacerbation rate. When compared to baseline, during biological therapy with benralizumab the number of exacerbations recorded in the previous year decreased from 8.00 ± 3.00 to 0.85 ± 1.53 (*p* < 0.0001) (Figure 1). The decrement of asthma exacerbation frequency due to benralizumab resulted to be statistically significant also in comparison to the pre-shift period (after treatment with omalizumab), when the mean number of exacerbations was 4.60 ± 3.10 (*p* < 0.01) (Figure 1). This result was paralleled by the reduction of the hospitalization rate, which decreased from the baseline value of 0.75 ± 0.71/year to 0.15 ± 0.36/year (*p* < 0.01) after treatment with omalizumab (pre-shift period), and zeroed after at least one year of add-on therapy with benralizumab (*p* < 0.001) (Figure 2). The mean number of daily SABA inhalations, used as needed rescue medications, was 0.10 ± 0.44 after therapy with benralizumab; this value was considerably lower than those detected at baseline (2.60 ± 1.69; *p* < 0.0001), and after add-on treatment with omalizumab (1.95 ± 1.82; *p* < 0.001), respectively (Figure 3). It was also possible to drastically reduce OCS intake, through a progressive lowering of OCS median dosage from baseline 11.25 mg (5.00–25.00) down to 5.00 mg (0.00–12.50; not significant) after at least 12 months of add-on therapy with omalizumab, and to 0.00 mg (0.00–0.00) (*p* < 0.001) after at least 12 months of add-on therapy with benralizumab, respectively (Figure 4). In particular, at baseline, 17 patients (85%) were on regular OCS treatment, who decreased to 14 (70%) after omalizumab therapy, and to 4 (20%) after benralizumab treatment, respectively. 

The median ACT score increased from 12.00 (7.00–15.00) at baseline to 20.00 (17.25–23.75) during treatment with benralizumab (*p* < 0.0001) (Figure 5). A significant rise of ACT score was experienced also when comparing post-benralizumab assessment to the result obtained before switching biological therapy, characterized by a median score of 15.00 (12.25–17.00) (*p* < 0.05). Moreover, in 9 patients also complaining of CRSwNP, the SNOT-22 score decreased from the mean baseline value of 58.44 ± 17.87 to 53.89 ± 15.93 after omalizumab treatment (*p* < 0.05), and to 43.44 ± 19.91 after benralizumab therapy (*p* < 0.01), respectively (Figure 6). We also detected a significant difference (*p* < 0.05) between the two SNOT-22 scores calculated after treatments with omalizumab and benralizumab, respectively (Figure 6). 

The mean pre-bronchodilator FEV_1_ enhanced from the baseline measurement of 1.71 ± 0.46 L at baseline to 2.18 ± 0.63 L after benralizumab therapy (*p* < 0.01). Mean pre-bronchodilator FEV_1_ also significantly increased with respect to pre-shift (post-omalizumab) assessment, characterized by a mean value of 1.81 ± 0.48 L (*p* < 0.05) (Figure 7). Finally, in addition to inducing the above clinical and functional effects, after at least 12 months of therapy benralizumab also zeroed the median blood eosinophil level, thus lowering it from the baseline count of 543.5 (360.0–1048) to 0.00 cells/μL (0.00–0.00) (*p* < 0.0001). The decrease of median blood eosinophil number was statistically significant even with respect to the time point of pre-switching biological therapy (*p* < 0.0001), corresponding to the end of treatment with omalizumab (Figure 8).

## 4. Discussion

The recent advances in the rapidly evolving scenario of the biological treatment of severe asthma have provided, during the last few years, an increasing number of valuable therapeutic options. Such a relevant progress makes it possible, in many cases, to tailor the biological therapy on the basis of the specific sub-phenotype expressed by each patient with severe type 2 asthma [20,23,55]. A careful phenotypic characterization relies on the combined clinical and functional evaluation, necessarily integrated by the measurement of useful biomarkers of type 2 asthma (serum IgE levels, blood eosinophil count, FeNO) [56]. The phenotype-driven first choice of a given biologic is often associated with a satisfactory therapeutic response. However, some patients with severe asthma do not achieve a sufficient control of their disease during the initial biological treatment, thus requiring a therapeutic shift [57]. This is especially true for patients treated with omalizumab, which has been for a long time the only approved anti-asthma monoclonal antibody [26]. Indeed, several studies have shown that allergic patients with severe asthma, who are not responsive to omalizumab, can be successfully switched to mepolizumab [34,35,36,37,38]. Nevertheless, to our knowledge scientific literature lacks publications referring to an eventual therapeutic shift from omalizumab to benralizumab. In regard to allergic patients with severe asthma previously uncontrolled by anti-IgE therapy, the present multi-center real-life experience clearly suggests that switching from omalizumab to benralizumab can induce significant improvements related to important clinical, functional, and biological aspects. This can be especially true for our severe asthmatic patients, who were characterized by a late-onset asthma, thus being more likely exposed to the risk of a poor response to omalizumab. In fact, real-life investigations have shown that the therapeutic response to omalizumab progressively decreases as age increases [58].

In particular, benralizumab significantly decreased the annual number of asthma exacerbations not only in comparison to baseline, but especially with respect to the effects of omalizumab, whose therapeutic action was quite modest thus not allowing a clinically relevant prevention of exacerbations, which was subsequently achieved after switching to benralizumab. Differently from omalizumab, benralizumab was indeed able to reduce asthma exacerbations much more than 50%, which is the threshold suggested by the British National Institute for Health and Care Excellence (NICE) as a reliable clinical goal to be pursued by anti-asthma therapies [59]. This impressive therapeutic effect of benralizumab is likely dependent on its anti-eosinophil activity, capable of zeroing blood eosinophil number, which on the contrary was not affected by omalizumab. In fact, a blood eosinophil count superior to 300–400 cells/μL is associated with a high risk of asthma exacerbations [60,61], and our patients had a median baseline value of blood eosinophils which was greater than 500 cells/μL. Hence, our results further corroborate the evidence that maintaining blood eosinophils at a low level decreases the risk of asthma exacerbations [62]. It can thus be argued that in some allergic patients with severe eosinophilic asthma like ours, the pharmacologic blockade of IgE is not sufficient to dramatically reduce disease exacerbations. Evidently, in such subjects the powerful IL-5R antagonism and the associated ADCC mechanism sustained by benralizumab are much more advantageous than anti-IgE therapy in providing an effective decrease in blood eosinophil count, and the consequent prevention of asthma exacerbations. Moreover, when compared to omalizumab treatment, the more effective benralizumab-induced decrement of asthma exacerbations made it possible for patients shifted to this drug to suppress their needs for hospitalization and OCS intake.

With respect to omalizumab treatment and in addition to the positive impact on the annual asthma exacerbation rate, benralizumab also elicited other clinical improvements, especially regarding symptom control. Indeed, our patients were characterized by a poor control of asthma symptoms, expressed by a quite low baseline ACT score, which modestly and not significantly enhanced after omalizumab treatment, whereas benralizumab induced an increase which resulted to be significant not only with respect to baseline, but also versus omalizumab therapy. In particular, after switching from omalizumab to benralizumab, our patients reached the crucial ACT score of 20, which reflects an adequate asthma control [63]. In real-life experiences like ours ACT is highly reliable as well as more practicable and easier to be performed when compared to asthma control questionnaires (ACQ), mostly used in RCT [64]. By remarkably improving the overall asthma control, benralizumab allowed a drastic reduction of SABA daily inhalations, utilized as rescue medication. Again, benralizumab significantly decreased SABA consumption not only with respect to baseline, but also in comparison to the slight and not significant effect produced by omalizumab. In addition to ameliorating asthma symptoms, benralizumab also improved the subjective symptoms related to upper airway diseases, as shown by the reduction of SNOT-22 score reported by our patients with severe asthma and nasal polyposis. The SNOT-22 score was further lowered by benralizumab when compared with the significant therapeutic action of omalizumab. These results are not very surprising, in consideration of recent similar data emerging from both RCT and real-world investigations, showing that both omalizumab and benralizumab are effective in improving the symptoms of nasal polyposis [48,49,65,66,67]. On the other hand, severe eosinophilic atopic asthma and allergic CRSwNP share common pathogenic mechanisms including the key roles played by IgE and IL-5-dependent airway eosinophilia [18,68,69]. Our patients with severe asthma and nasal polyposis responded very well to benralizumab, not differently than those without nasal polyps.

In regard to lung function, at baseline our patients were characterized by a relevant airflow limitation (mean pre-bronchodilator FEV_1_: 62% pred). The initial treatment with omalizumab induced a scanty and not significant FEV_1_ change, which was overwhelmed by the subsequent therapy with benralizumab, responsible for a significant FEV_1_ increase amounting after 1 year to more than 400 mL with respect to baseline, and to more than 300 mL when compared to the end of omalizumab treatment (pre-switching), respectively. Such a remarkable improvement of bronchial obstruction, provided by benralizumab, is probably due to the powerful anti-eosinophil activity of this drug. Although the relationship between blood and sputum eosinophil levels is still debated, and with regard to this context there is disagreement in scientific literature, a recent reliable study has clearly shown that blood eosinophils are an excellent predictor of sputum eosinophilia in patients with uncontrolled asthma [70]. Therefore, because blood eosinophil numbers reflect the entity of airway eosinophilia, it is likely that eosinophilic inflammation leads to the development of airflow limitation. This pathogenic circuit can thus be effectively interrupted by benralizumab, but not by omalizumab, which in our patients did not affect at all the level of blood eosinophils.

Taken together, the results of the present real-life study strongly suggest that a subset of allergic patients with severe eosinophilic asthma, unresponsive to an initial add-on treatment with omalizumab, can greatly benefit from a subsequent biological therapy with benralizumab. This therapeutic shift was allowed by the phenotypic features of our patients, belonging to a group of subjects who express high levels of multiple biomarkers of type 2 asthma, including serum IgE and blood eosinophils [71]. Of course, the primary goal of clinicians managing severe asthma should be to initially pursue the best possible therapeutic choice for their patients. However, when some asthmatic patients experience residual symptoms and exacerbations despite their first access to a phenotype-based biological therapy, physicians must investigate eventual alternative approaches. This is particularly true in regard to the frequent overlap between allergic and eosinophilic traits, which often shapes a multidimensional eligibility of severe asthmatic patients to anti-IgE treatment, as well as to anti-IL-5, anti-IL-5 receptor, and anti-IL-4/IL-13 receptor therapies [72,73]. Within such a context, blood eosinophil counts can play an important predictive role. Indeed, high blood eosinophil numbers, but not serum IgE levels, are associated with a high risk of asthma exacerbations in patients with severe, uncontrolled disease [74]. These considerations can help to explain our current data referring to allergic patients with severe eosinophilic asthma switched from omalizumab to benralizumab, which has been shown to be very effective in such subjects independently of serum IgE concentrations [74]. Whilst elevated blood eosinophil counts are predictive of benralizumab efficacy, serum IgE levels do not predict the response to omalizumab [74,75]. Hence, a plausible reading key for our results, obtained in asthmatic patients with quite elevated blood eosinophil numbers, may imply that omalizumab-resistant eosinophilic inflammation can be adequately controlled by benralizumab. It can thus be hypothesized that in our allergic patients the main pathobiologic mechanism, underlying severe eosinophilic asthma, originated from an excessive local and systemic production of IL-5, responsible for high counts of eosinophils in both airways and blood. This IL-5-dependent eosinophilic inflammation can be suppressed by benralizumab, but not by omalizumab. Such a speculation could be extended to both upper and lower airways for our patients complaining of severe eosinophilic asthma and nasal polyposis, highly susceptible to the add-on biological therapy with benralizumab [48,49,65,66]. Among the comorbidities of severe asthma, nasal polyposis is indeed one of the most reliable predictors of an effective therapeutic response to benralizumab [76]. 

Of course, omalizumab remains the first drug selection for uncontrolled atopic asthmatics [14,20], whose clinical history is clearly characterized by a pivotal role played by allergic sensitization. Based on this consideration, we initially chose omalizumab as add-on biological therapy for our patients with severe atopic asthma, being convinced that IgE-mediated allergic pathways were the main mechanisms responsible for both clinical and functional features. However, in such subjects a close attention should also be paid to blood eosinophil count. Indeed, in case of the eventual concomitance of a relevant blood eosinophilia, the therapeutic orientation could deviate from omalizumab towards mepolizumab, reslizumab, or benralizumab. Anyway, if a phenotypic-driven prescription of omalizumab does not yield satisfactory clinical results within a year of treatment, a theapeutic switch to another biologic should be considered.

In conclusion, we think that the main strength of our real-life experience is based on the demonstration of the effectiveness of the pharmacologic shift from omalizumab to benralizumab, detected in allergic patients with severe eosinophilic asthma. These results are very interesting, because the effects of the above therapeutic switching are largely unexplored in atopic subjects with difficult-to-treat type 2 asthma. On the other hand, similar to other real-world investigations, our present study has some limitations including the relatively low number of enrolled patients, as well as the lack of randomization and placebo control. 

## Figures and Tables

**Figure 1 biomedicines-09-01822-f001:**
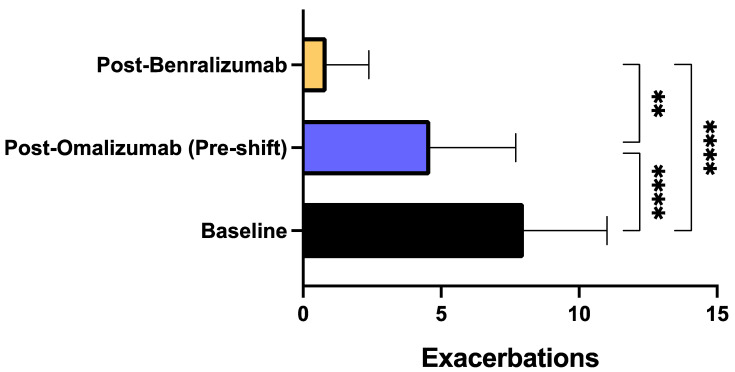
Effects of omalizumab and benralizumab on the annual number of asthma exacerbations. Comparisons were made between baseline and omalizumab, baseline and benralizumab, and omalizumab and benralizumab, respectively. ** *p* < 0.01; **** *p* < 0.0001.

**Figure 2 biomedicines-09-01822-f002:**
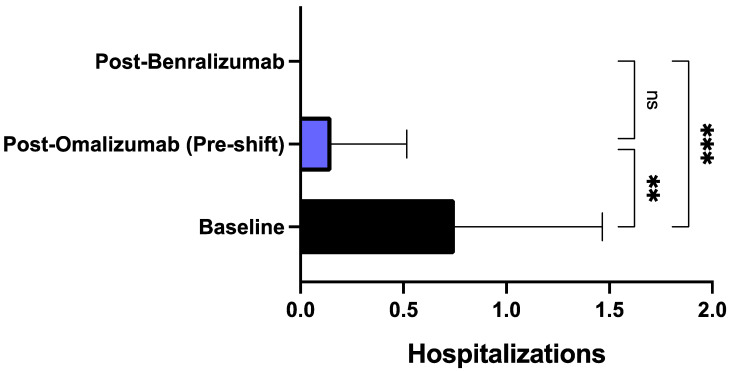
Effects of omalizumab and benralizumab on the annual number of hospitalizations for asthma exacerbations. Comparisons were made between baseline and omalizumab, baseline and benralizumab, and omalizumab and benralizumab, respectively. ** *p* < 0.01; *** *p* < 0.001. ns: not significant.

**Figure 3 biomedicines-09-01822-f003:**
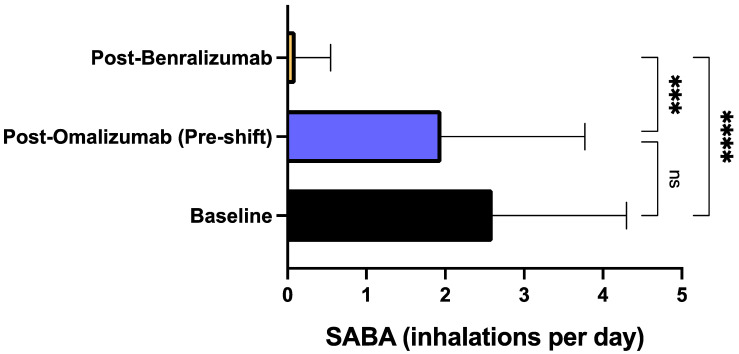
Effects of omalizumab and benralizumab on the daily number of SABA inhalations. Comparisons were made between baseline and omalizumab, baseline and benralizumab, and omalizumab and benralizumab, respectively. *** *p* < 0.001; **** *p* < 0.0001. ns: not significant.

**Figure 4 biomedicines-09-01822-f004:**
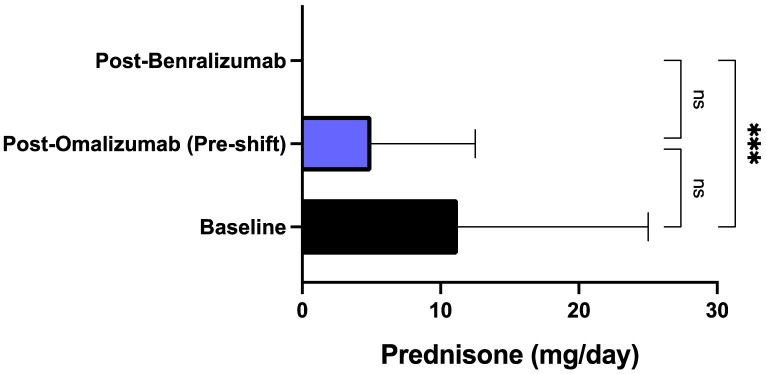
Effects of omalizumab and benralizumab on the daily prednisone intake. Comparisons were made between baseline and omalizumab, baseline and benralizumab, and omalizumab and benralizumab, respectively. *** *p* < 0.001. ns: not significant.

**Figure 5 biomedicines-09-01822-f005:**
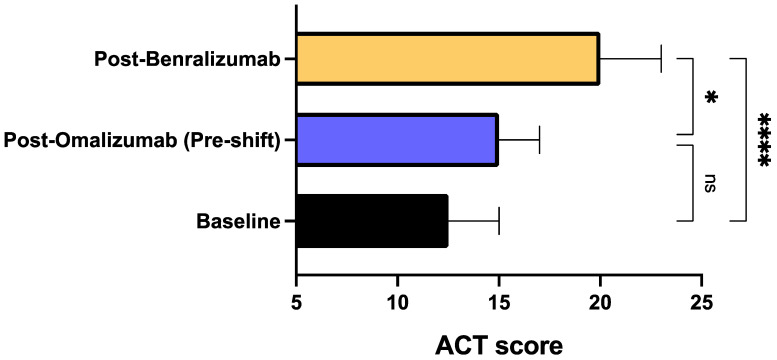
Effects of omalizumab and benralizumab on ACT score. Comparisons were made between baseline and omalizumab, baseline and benralizumab, and omalizumab and benralizumab, respectively. * *p* < 0.05; **** *p* < 0.0001. ns: not significant.

**Figure 6 biomedicines-09-01822-f006:**
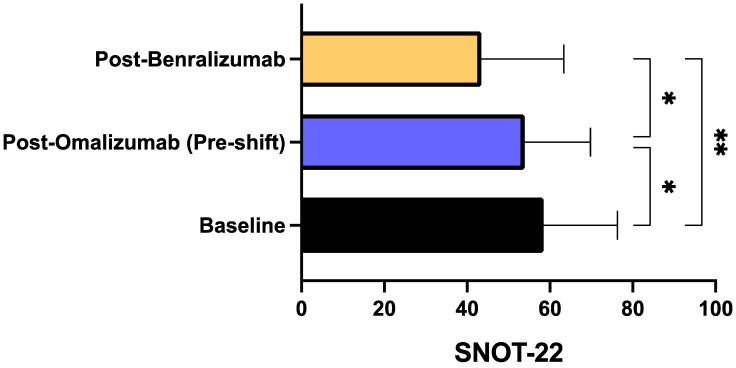
Effects of omalizumab and benralizumab on SNOT-22 score. Comparisons were made between baseline and omalizumab, baseline and benralizumab, and omalizumab and benralizumab, respectively. * *p* < 0.05; ** *p* < 0.01.

**Figure 7 biomedicines-09-01822-f007:**
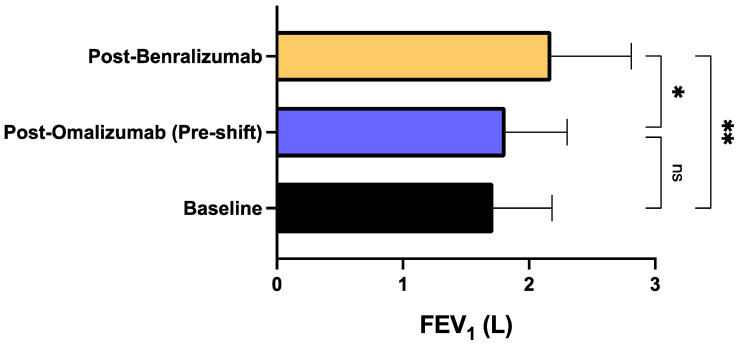
Effects of omalizumab and benralizumab on FEV_1_. Comparisons were made between baseline and omalizumab, baseline and benralizumab, and omalizumab and benralizumab, respectively. * *p* < 0.05; ** *p* < 0.01. ns: not significant.

**Figure 8 biomedicines-09-01822-f008:**
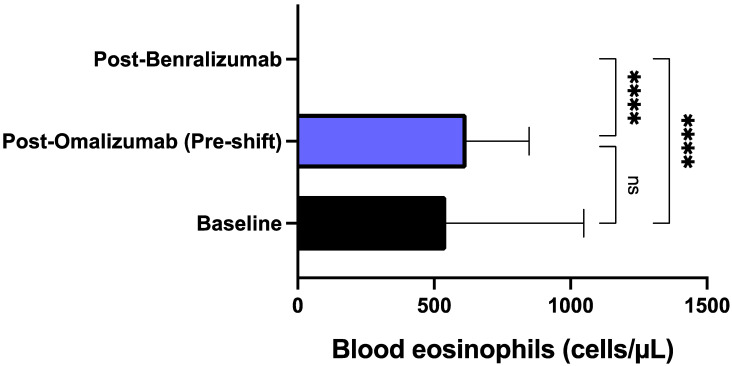
Effects of omalizumab and benralizumab on blood eosinophil counts. Comparisons were made between baseline and omalizumab, baseline and benralizumab, and omalizumab and benralizumab, respectively. **** *p* < 0.0001. ns: not significant.

**Table 1 biomedicines-09-01822-t001:** Baseline patient features.

Characteristics	Total Population (N = 20)
Female gender, N (%)	13 (65)
Male gender, N (%)	7 (35)
Age, mean (±SD), years	52.85 ± 9.39
Asthma onset age, mean (±SD), years	31.40 ± 14.34
Duration of asthma, mean (±SD), years	21.85 ± 15.90
BMI, mean (±SD), % predicted	23.47 ± 3.54
FEV_1_, mean (±SD), % predicted	62.47 ± 13.84
FEV_1_/FVC, mean (±SD), %	61.15 ± 11.65
Blood eosinophils, median value (IQR), cells/μL	543.5 (360.0–1048)
Total serum IgE, median value (IQR), IU/mL	274.5 (198.8–412.8)
Gastro-esophageal reflux disease, N (%)	11 (55)
Chronic rhinosinusitis with nasal polyps, N (%)	9 (45)
Bronchiectasis, N (%)	7 (35)
Atopic dermatitis, N (%)	3 (15)
Obstructive sleep apnea syndrome, N (%)	2 (10)
On treatment with ICS/LABA, N (%)	20 (100)
On treatment with LAMA, N (%)	15 (75)
On treatment with LTRA drugs, N (%)	14 (70)

Abbreviations: BMI, body mass index; FEV_1_, forced expiratory volume in the first second; FVC, forced vital capacity; ICS, inhaled corticosteroid; LABA, long-acting β_2_-adrenergic agonist; LAMA, long-acting muscarinic receptor antagonist; LTRA, leukotriene receptor antagonist; SD, standard deviation; IQR, interquartile range.

## Data Availability

The data presented in this study are available on request from the corresponding author.

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
