# Peer review of "Switch from Omalizumab to Benralizumab in Allergic Patients with Severe Eosinophilic Asthma: A Real-Life Experience from Southern Italy"

_biomedicines, 2021, doi:10.3390/biomedicines9121822_

Round 1

Reviewer 1 Report

  1. Interesting study and manuscript
  2. Suggestion to make more clear how many patients initially treated with omalizumab had  blood eosinophilia and to separately characterize this subgroup , due to expected better outcome  to benralizumab.
  3. Is it the same who associated CRSwNP ?
  4. Line 121 : Omalizumab failure was  defined after 12 months...is this time period  based on any guidelines recommendation ?
  5. Due to hight cost of biologicals , we should also consider the cost - effectiveness aspects of this indication and probably restrict the initial treatment evaluation to 3-6 months , before  switching  to another biological . Please add your comment.
  6. Correlation between blood and airways eosinophilia is still questionable in some studies , the authors comment on this aspect might be useful.
  7. Suggested reference : Real-life  benefit of Omalizumab in improving control of bronchial asthma during COVID-19 pandemic. Case report. Polliana Mihaela Leru, Anton Vlad Florin. Cureus 2021.

Author Response

REVIEWER 1

1. Interesting study and manuscript.

We would like to thank very much this reviewer for his/her appreciation of our manuscript.

2. Suggestion to make more clear how many patients initially treated with omalizumab had blood eosinophilia and to separately characterize this subgroup, due to expected better outcome to benralizumab.

This point has been addressed in our revised manuscript (page 3, lines 115-117).

3. Is it the same who associated CRSwNP ?

This point has been addressed in our revised manuscript (page 3, lines 115-117).

4. Line 121: Omalizumab failure was  defined after 12 months...is this time period  based on any guidelines recommendation ?

This point has been addressed in our revised manuscript (page 3, lines 125-130).

5. Due to high cost of biologicals , we should also consider the cost - effectiveness aspects of this indication and probably restrict the initial treatment evaluation to 3-6 months, before  switching  to another biological. Please add your comment.

This point has been addressed in our revised manuscript (page 3, lines 125-130).

6. Correlation between blood and airways eosinophilia is still questionable in some studies, the authors comment on this aspect might be useful.

This point has been addressed in our revised manuscript (page 12, lines 339-343).

7. Suggested reference: Real-life benefit of Omalizumab in improving control of bronchial asthma during COVID-19 pandemic. Case report. Polliana Mihaela Leru, Anton Vlad Florin. Cureus 2021.

This point has been addressed in our revised manuscript (page 2, lines 78-80), and the suggested reference has been included (ref. n. 33).

Reviewer 2 Report

The authors addressed the research question if treatment of omalizumab non responsive patients with benralizumab is beneficial. The authors run a clinical studies, with standard classification and so far standard treatment of patients.

The manuscript fit together very well. It is easy to read and to understand. It provides a meaningful introduction and hypothesis. The method was according to standard procedures. It clearly describes the current status of knowledge and methods, what was done and why. The results are interpreted. The manuscript is written well. Although it is only a very small clinical studies, the findings might be interesting for others.

Author Response

REVIEWER 2

The authors addressed the research question if treatment of omalizumab non responsive patients with benralizumab is beneficial. The authors run a clinical study, with standard classification and so far standard treatment of patients.

The manuscript fits together very well. It is easy to read and to understand. It provides a meaningful introduction and hypothesis. The method was according to standard procedures. It clearly describes the current status of knowledge and methods, what was done and why. The results are interpreted. The manuscript is written well. Although it is only a very small clinical study, the findings might be interesting for others.

We would like to thank very much this reviewer for his/her appreciation of our manuscript.

Reviewer 3 Report

The authors of this study evaluate in a real-life setting the response of severe allergic eosinophilic asthma after switch of treatment from omalizumab to benralizumab. They found that such a shift was associated with a significant improvement of outcomes including exacerbation rate, hospitalization rate, OCS use, lung function, asthma control and SABA use.

This is a well-written and interesting study, however some concerns are raised.

Major comments

  1. Abstract, lines 37-39: “The results of this real-life study suggest that the pharmacologic shift from omalizumab to benralizumab can be a valuable therapeutic approach for allergic patients with severe eosinophilic asthma, eventually associated with nasal polyposis, not adequately controlled by anti-IgE treatment”. The attribution of this valuable approach is not confined to those patients with nasal polyposis as this subgroup included 9/20 patients and in the manuscript there is no comparison of the switch effect between those with and without nasal polyps. Accordingly, I suggest that “eventually associated with nasal polyposis” should be deleted from the sentence.
  2. Patients and methods, line 122: The suggested by GINA severe asthma pocket guide time for evaluation of response to omalizumab is 16 weeks. In this study, the unsatisfactory clinical response to omalizumab was defined after at least 12 months of add-on therapy. What made the authors to prolong the time assessment of response to omalizumab?
  3. Patients and methods, lines 122-123: The unsatisfactory clinical response to omalizumab was defined as lack of effectiveness, featured by recurrent asthma exacerbations and/or uncontrolled symptoms. The authors should describe more accurately the criteria used for non-response. For example there was a decrease of exacerbation rate from 8/year, to 4,6/year. Although 4,6 exacerbations/year is not satisfactory still it is nearly a 50% decrease.
  4. Patients and methods, line 141: The authors report that the mean age of asthma onset age was 31.40 ± 14.34 years. This is more compatible with late-onset than early-onset asthma that usually starts early in life and is more often allergic. The authors should comment in the discussion.
  5. There was a subgroup of 9/20 (45%) severe asthma patients with nasal polyps. Did this group respond more favorably to benralizumab compared to those without nasal polyps?
  6. Discussion, lines 275-277: “Conversely, the persistence of unchanged high blood eosinophil numbers, which we detected during the initial treatment with omalizumab, can explain the inability of this drug to lower asthma exacerbations to a clinically relevant extent”. The study of Hanania et al, showed a better response to omalizumab if blood eosinophils were >280/μL but the STELLAIR study showed that omalizumab was effective regardless of blood eosinophil number. Accordingly, absolute blood eosinophil count is not a prognostic biomarker for response to omalizumab. Blood eosinophil count is not a marker for assessing omalizumab efficacy as well. Meaning that the lack of decrease of blood eosinophils is not indicative of a lack of response to omalizumab. The authors should rephrase.
  7. A main concern arising in this study (and also other similar studies) is the initial choice of biologic treatment. It is clear that severe asthma patients with the overlap phenotype are eligible for either anti-IgE or anti-IL-5/5R treatment. But it is important to choose at first the most likely for the patient to respond. Accordingly, the question arises whether these 20 patients with the overlap phenotype who had late-onset asthma, with a very high number of exacerbations (8/year) and a baseline absolute eosinophil count of 543/μL and nasal polyps in almost half of them the initial choice of omalizumab was the proper one or they should have received benralizumab (or mepolizumab) at first. It is of utmost importance to comment on the rationale for choosing omalizumab as the initial treatment.

Minor comments

  1. Introduction, line 62: “Indeed, elevated concentrations of serum IgE favour the prescription of anti IgE-treatment …”. This is not true. Levels of IgE are considered for defining the dosage of omalizumab. The allergic phenotype defined by positive SPT to a perennial allergen is what favors omalizumab in a severe asthmatic. The authors should rephrase.
  2. Patients and methods, line 117: “… within the range of 30-1500IU/mL”. The range is 30-700IU/mL for adults. It is up to 1500IU/mL for children.
  3. How many patients were on regular OCS at baseline and after omalizumab treatment and how many stopped OCS after benralizumab? Data on median OCS dose are only given. Also the median dose of OCS after omalizumab is not mentioned (although not significant from baseline it seems reduced in the figure).
  4. In this study, SNOT-22 was significantly lower after omalizumab compared to baseline and was further reduced by benralizumab. Omalizumab is also indicated for nasal polyps (according to the recent studies POLYP-1 and POLYP-2) and this should be commented in the discussion.
  5. Discussion, line 316: “In fact, it is well known that blood eosinophil count is inversely correlated to FEV1% pred.” This is based on an old study (ref 66, 1998) and the inverse correlation of blood eosinophil count and lung function has not been confirmed by more recent studies in severe asthma. I suggest that the authors delete the sentence.

Author Response

REVIEWER 3

The authors of this study evaluate in a real-life setting the response of severe allergic eosinophilic asthma after switch of treatment from omalizumab to benralizumab. They found that such a shift was associated with a significant improvement of outcomes including exacerbation rate, hospitalization rate, OCS use, lung function, asthma control and SABA use.

This is a well-written and interesting study, however some concerns are raised.

We would like to thank very much this reviewer for his/her appreciation of our manuscript.

Major comments

1. Abstract, lines 37-39: “The results of this real-life study suggest that the pharmacologic shift from omalizumab to benralizumab can be a valuable therapeutic approach for allergic patients with severe eosinophilic asthma, eventually associated with nasal polyposis, not adequately controlled by anti-IgE treatment”. The attribution of this valuable approach is not confined to those patients with nasal polyposis as this subgroup included 9/20 patients and in the manuscript there is no comparison of the switch effect between those with and without nasal polyps. Accordingly, I suggest that “eventually associated with nasal polyposis” should be deleted from the sentence.

This deletion has been made in our revised manuscript (page 1, line 39).

2. Patients and methods, line 122: The suggested by GINA severe asthma pocket guide time for evaluation of response to omalizumab is 16 weeks. In this study, the unsatisfactory clinical response to omalizumab was defined after at least 12 months of add-on therapy. What made the authors to prolong the time assessment of response to omalizumab?

This point has been addressed in our revised manuscript (page 3, lines 125-130).

3. Patients and methods, lines 122-123: The unsatisfactory clinical response to omalizumab was defined as lack of effectiveness, featured by recurrent asthma exacerbations and/or uncontrolled symptoms. The authors should describe more accurately the criteria used for non-response. For example there was a decrease of exacerbation rate from 8/year, to 4,6/year. Although 4,6 exacerbations/year is not satisfactory still it is nearly a 50% decrease.

This point has been addressed in our revised manuscript (page 3, lines 132-136).

4. Patients and methods, line 141: The authors report that the mean age of asthma onset was 31.40 ± 14.34 years. This is more compatible with late-onset than early-onset asthma that usually starts early in life and is more often allergic. The authors should comment in the discussion.

This point has been addressed in our revised manuscript (page 11, lines 275-279).

5. There was a subgroup of 9/20 (45%) severe asthma patients with nasal polyps. Did this group respond more favorably to benralizumab compared to those without nasal polyps?

This point has been addressed in our revised manuscript (page 12, lines 328-330).

6. Discussion, lines 275-277: “Conversely, the persistence of unchanged high blood eosinophil numbers, which we detected during the initial treatment with omalizumab, can explain the inability of this drug to lower asthma exacerbations to a clinically relevant extent”. The study of Hanania et al, showed a better response to omalizumab if blood eosinophils were >280/μL but the STELLAIR study showed that omalizumab was effective regardless of blood eosinophil number. Accordingly, absolute blood eosinophil count is not a prognostic biomarker for response to omalizumab. Blood eosinophil count is not a marker for assessing omalizumab efficacy as well. Meaning that the lack of decrease of blood eosinophils is not indicative of a lack of response to omalizumab. The authors should rephrase.

This sentence has been deleted in our revised manuscript (page 11, lines 293-296).

7. A main concern arising in this study (and also other similar studies) is the initial choice of biologic treatment. It is clear that severe asthma patients with the overlap phenotype are eligible for either anti-IgE or anti-IL-5/5R treatment. But it is important to choose at first the most likely for the patient to respond. Accordingly, the question arises whether these 20 patients with the overlap phenotype who had late-onset asthma, with a very high number of exacerbations (8/year) and a baseline absolute eosinophil count of 543/μL and nasal polyps in almost half of them the initial choice of omalizumab was the proper one or they should have received benralizumab (or mepolizumab) at first. It is of utmost importance to comment on the rationale for choosing omalizumab as the initial treatment.

This point has been addressed in our revised manuscript (page 13, lines 383-386).

Minor comments

1. Introduction, line 62: “Indeed, elevated concentrations of serum IgE favour the prescription of anti IgE-treatment …”. This is not true. Levels of IgE are considered for defining the dosage of omalizumab. The allergic phenotype defined by positive SPT to a perennial allergen is what favors omalizumab in a severe asthmatic. The authors should rephrase.

This point has been addressed in our revised manuscript (page 2, lines 62-63).

2. Patients and methods, line 117: “… within the range of 30-1500 IU/mL”. The range is 30-700 IU/mL for adults. It is up to 1500 IU/mL for children.

This point has been addressed in our revised manuscript (page 3, line 121).

3. How many patients were on regular OCS at baseline and after omalizumab treatment and how many stopped OCS after benralizumab? Data on median OCS dose are only given. Also the median dose of OCS after omalizumab is not mentioned (although not significant from baseline it seems reduced in the figure).

This point has been addressed in our revised manuscript (page 6, lines 194-199).

4. In this study, SNOT-22 was significantly lower after omalizumab compared to baseline and was further reduced by benralizumab. Omalizumab is also indicated for nasal polyps (according to the recent studies POLYP-1 and POLYP-2) and this should be commented in the discussion.

This point has been addressed in our revised manuscript (page 12, lines 323-326), and the reference regarding the studies POLYP-1 and POLYP-2 has been included (ref. n. 67).

5. Discussion, line 316: “In fact, it is well known that blood eosinophil count is inversely correlated to FEV1% pred.” This is based on an old study (ref 66, 1998) and the inverse correlation of blood eosinophil count and lung function has not been confirmed by more recent studies in severe asthma. I suggest that the authors delete the sentence.

This sentence and the relative reference have been deleted in our revised manuscript (page 12, lines 338-339).